# Integrative Wound-Healing Effects of *Clinacanthus nutans* Extract and Schaftoside Through Anti-Inflammatory, Endothelial-Protective, and Antiviral Mechanisms

**DOI:** 10.3390/ijms26136029

**Published:** 2025-06-23

**Authors:** Nipitpawn Limpanich, Pattarasuda Chayapakdee, Kullanun Mekawan, Saruda Thongyim, Rujipas Yongsawas, Phanuwit Khamwong, Yingmanee Tragoolpua, Thida Kaewkod, Siriphorn Jangsutthivorawat, Jarunee Jungklang, Usawadee Chanasut, Angkhana Inta, Phatchawan Arjinajarn, Aussara Panya, Hataichanok Pandith

**Affiliations:** 1Department of Biology, Faculty of Science, Chiang Mai University, Mueang Chiang Mai, Chiang Mai 50200, Thailand; npplpnch123@gmail.com (N.L.); pattarasuda.c@cmu.ac.th (P.C.); kullanun_me@cmu.ac.th (K.M.); saruda_th@cmu.ac.th (S.T.); r.yongsawas@gmail.com (R.Y.); panuwit_ka@cmu.ac.th (P.K.); yingmanee.t@cmu.ac.th (Y.T.); thida.kaewkod@cmu.ac.th (T.K.); siriphorn.biocmu@gmail.com (S.J.); jarunee.j@cmu.ac.th (J.J.); usawadee.chanasut@cmu.ac.th (U.C.); aungkanainta@hotmail.com (A.I.); arjinajarn_jin@hotmail.com (P.A.); 2Office of Research Administration, Chiang Mai University, Chiang Mai 50200, Thailand

**Keywords:** *Clinacanthus nutans*, schaftoside, wound healing, COX-2 inhibition, endothelial protection, inflammation, HSV-2

## Abstract

*Clinacanthus nutans* (Burm.f.) Lindau is a Southeast Asian medicinal plant traditionally used for treating skin inflammation and infections. This study evaluated its wound-healing potential through anti-inflammatory, cytoprotective, and antiviral mechanisms. HPLC-DAD analysis identified schaftoside as the major flavonoid in the 95% ethanolic leaf extract. In the lipopolysaccharide (LPS)-stimulated murine macrophage cell line (RAW 264.7), both *C*. *nutans* extract (5 and 50 μg/mL) and its flavonoid schaftoside (5 and 20 μg/mL) significantly downregulated the expression of pro-inflammatory genes, including *cyclooxygenase-2* (*COX-2*), *inducible nitric oxide synthase* (*iNOS*), *interleukin-6* (*IL-6*), *tumor necrosis factor-alpha* (*TNF-α*), and *prostaglandin E*_2_ (*PGE*_2_), under both pre-treatment and post-treatment conditions. ELISA confirmed dose-dependent inhibition of human COX-2 enzymatic activity, reaching up to 99.3% with the extract and 86.9% with schaftoside. In the endothelial cell models (CCL-209), the extract exhibited low cytotoxicity and effectively protected cells from LPS-induced apoptosis, preserving vascular integrity critical to tissue regeneration. Antiviral assays demonstrated suppression of HSV-2 replication, particularly during early infection, which may help prevent infection-related delays in wound healing. Collectively, these findings suggest that *C. nutans* and schaftoside promote wound repair by attenuating inflammatory responses, supporting endothelial survival, and controlling viral reactivation. These multifunctional properties highlight their potential as natural therapeutic agents for enhancing wound-healing outcomes.

## 1. Introduction

Wound healing depends heavily on the inflammatory phase, which initiates repair through immune cell interactions, cytokines, and blood vessels. Disruptions in inflammation—caused by infection, oxidative stress, or chronic conditions—can hinder healing and lead to chronic wounds or scarring. Recently, plant-based therapeutics are gaining attention for their multi-targeted effects, biocompatibility, and low side effects in supporting this process [1,2].

*Clinacanthus nutans* Lindau (commonly known in Thailand as “Phaya Yo”) is a small medicinal shrub in the Acanthaceae family, widely distributed and traditionally cultivated across southeast Asia, particularly in Thailand. For generations, *C. nutans* has been used in Thai traditional medicine to alleviate various dermatological and inflammatory conditions, including insect bites, snake bites, rashes, and viral lesions caused by herpes simplex virus (HSV) and varicella-zoster virus (VZV) [2,3,4]. The ethnobotanical knowledge of *C. nutans* has been supported by modern pharmacological evidence demonstrating its therapeutic potential, particularly its anti-inflammatory, antiviral, and anti-apoptotic activities, both in vitro and in vivo [5,6]. These properties are highly relevant to wound repair, especially in modulating inflammatory responses.

Lipopolysaccharide (LPS)-induced Toll-like receptor 4 (TLR4) signaling in macrophages leads to reduced activation of mitogen-activated protein kinases (MAPKs), including p38, extracellular signal-regulated kinase 1/2 (ERK1/2), and c-Jun N-terminal kinase 1/2 (JNK1/2), and the transcription factors nuclear factor-kappa B (NF-κB) and interferon regulatory factor 3 (IRF3). These pathways ultimately upregulate the expression of pro-inflammatory mediators, including inducible nitric oxide synthase (iNOS), cyclooxygenase-2 (COX-2), tumor necrosis factor-alpha (TNF-α), interleukin-6 (IL-6), nitric oxide (NO), and prostaglandin E_2_ (PGE_2_), which contribute to tissue damage and delayed wound healing [2,7,8,9]. In various endothelial cell models, natural products—including extracts from *C. nutans*—have demonstrated protective effects against inflammation-induced endothelial injury and apoptosis, helping to maintain vascular integrity under stress conditions such as lipopolysaccharide (LPS) exposure or oxidative insults [2,10,11]. Moreover, in the context of viral infections such as herpes simplex virus type 2 (HSV-2), cellular entry depends on viral glycoproteins that mediate host cell attachment and membrane fusion. Disruption of these glycoproteins can effectively prevent viral penetration and subsequent replication [12]. Collectively, modulation of immune activation, preservation of endothelial integrity, and interference with viral entry mechanisms converge to create a tissue microenvironment that favors fibroblast recruitment, proliferation, and extracellular matrix remodeling—key processes in wound healing and tissue regeneration. These findings suggest that *C. nutans* modulates both immune and endothelial cellular responses, contributing to a coordinated wound-healing process. The combined anti-inflammatory and vasculoprotective actions are likely to establish a regenerative microenvironment favorable to fibroblast recruitment, activation, and extracellular matrix remodeling—key events during the proliferative and remodeling phases of wound repair [13,14].

Chemotypic variation studies have revealed that *C. nutans* specimens from northern Thailand—particularly from provinces such as Lampang, Chiang Mai, and Phayao—exhibit higher phenolic content and enhanced antioxidant, anti-apoptotic, and antibacterial activities relative to those from other regions [15]. This study specifically investigates *C. nutans* leaves collected from Lampang, focusing on schaftoside, a stable C-glycosyl flavone with notable pharmacological properties [6,16]. While other flavonoids such as vitexin, isovitexin, orientin, and isoorientin are also present, schaftoside has consistently emerged as the major bioactive component of interest due to its chemical stability—attributed to its C–C glycosidic bond—and potential as a chemotaxonomic marker [6,10,17,18,19].

This work aims to elucidate how *C. nutans* extract and schaftoside modulate inflammatory mediators, preserve endothelial viability under stress, and attenuate viral effects that may hinder wound resolution. Understanding these mechanisms will contribute to the rational development of phytotherapeutics for dermal and mucosal wound management.

## 2. Results

### 2.1. Optimization and Validation of the High-Performance Liquid Chromatography Method with Diode-Array Detection Method (HPLC-DAD) for Schaftoside Quantification

A HPLC-DAD was developed and validated for the quantification of schaftoside in ethanolic extracts of *C. nutans* leaves. The method demonstrated excellent linearity over the concentration range of 0.78–50 μg/mL, with a correlation coefficient (R^2^) of 0.9999 (Figure 1A). Chromatographic analysis of the schaftoside standard (100 μg/mL) showed a retention time of 12.131 ± 0.0042 min and a symmetric peak profile (Figure 1B). The method exhibited high precision, with %RSD values of 0.2542% (intraday) and 0.6940% (interday), and high accuracy, with recovery rates ranging from 96.978% to 98.714% (Figure 1C). Sensitivity parameters indicated an LOD of 0.5635 μg/mL and an LOQ of 1.7077 μg/mL. Application of the method to the plant extract revealed a schaftoside concentration of 3.6475 μg/mL, equivalent to 121.5838 mg per 100 g of fresh leaves (Figure 1D). These findings confirm that the developed method is robust, accurate, precise, and suitable for routine quantification of schaftoside in *C. nutans* and other botanical matrices.

### 2.2. Anti-Inflammatory Effects of C. nutans Extract and Schaftoside

To investigate the anti-inflammatory potential of *C. nutans* and its bioactive flavonoid, schaftoside, a series of cytotoxicity, Western blot, and RT-PCR analyses were conducted in LPS-stimulated RAW 264.7 macrophages—an established model for inflammation. The fresh leaves of *C. nutans* were further extracted with 95% ethanol.

#### 2.2.1. Cytotoxicity and Dose Selection (Figure 2A,B)

To assess cytotoxicity, RAW 264.7 cells were treated with various concentrations of 95% ethanolic *C. nutans* extract and its bioactive compound, schaftoside. As shown in Figure 2A, cell viability assays revealed that *C. nutans* extract was non-toxic up to 250 μg/mL, with viability remaining above 85%. Similarly, schaftoside at concentrations of 10 and 20 μg/mL did not induce cytotoxicity (Figure 2B). These concentrations were therefore selected as sublethal doses for further anti-inflammatory testing.

**Figure 2 ijms-26-06029-f002:**
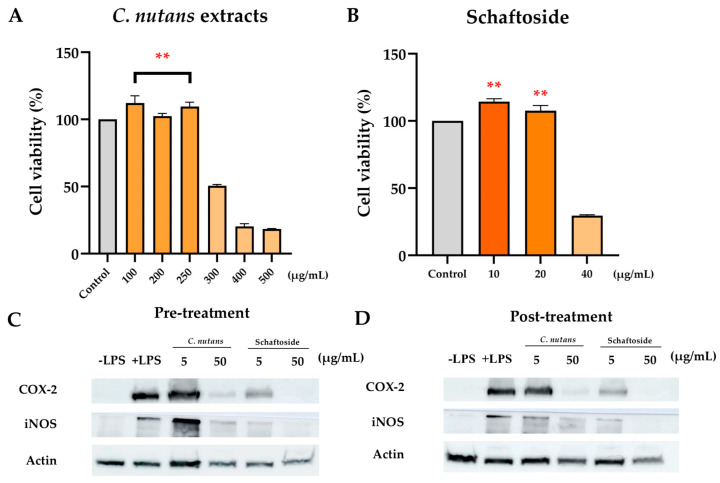
(**A**) The cytotoxicity of *C. nutans* extract and (**B**) its bioactive compound, schaftoside, was evaluated in RAW 264.7 cells using the CellTiter 96^®^ AQueous One Solution Cell Proliferation Assay. Results are presented as mean ± SD from four replicates. To assess anti-inflammatory effects, Western blot analysis was performed on LPS-stimulated RAW 264.7 cells. In the pre-treatment model (**C**), cells were incubated with 95% ethanolic *C. nutans* extract or schaftoside for 1 h prior to stimulation with 100 ng/mL LPS for 12 h. In the post-treatment model (**D**), cells were first exposed to LPS for 1 h, followed by treatment with the extract or schaftoside for 12 h. Expression levels of COX-2 and iNOS proteins were examined, with actin used as a loading control. Asterisks indicate significant differences compared to the control group (** *p* < 0.01).

#### 2.2.2. Suppression of Inflammatory Proteins (Figure 2C,D)

Western blot analysis was used to assess protein levels of COX-2 and iNOS, two key pro-inflammatory enzymes, under both pre-treatment and post-treatment conditions: In pre-treatment (cells treated with plant compounds for 1 h prior to stimulation with 100 ng/mL before LPS stimulation for 12 h), both *C. nutans* extract at 50 µg/mL and schaftoside at 20 µg/mL significantly reduced COX-2 and iNOS expression, as indicated by diminished band intensities. In post-treatment (cells treated with compounds for 12 h after LPS exposure for 1 h), similar inhibitory trends were observed, although slightly weaker at lower concentrations of 5 µg/mL of schaftoside. Actin expression remained stable across all conditions, confirming equal protein loading. These findings highlight the dose-dependent suppression of inflammation-related proteins, with both treatments most effective at higher doses (50 μg/mL for extract and 20 μg/mL for schaftoside).

#### 2.2.3. Inhibitory Effects of *C. nutans* Extract and Schaftoside on Human Cox-2 Enzyme Activity

To further evaluate the anti-inflammatory potential of *C. nutans* and its major flavonoid schaftoside, their effects on human cyclooxygenase-2 (COX-2) enzyme activity were assessed using a commercial COX (human) inhibitor screening kit. As shown in Figure 3A, *C. nutans* extract exhibited dose-dependent inhibition of COX-2 activity. At a concentration of 0.5 mg/mL, the extract inhibited COX-2 of 82.3%, with increasing inhibition observed at higher concentrations, reaching a maximum of 99.3% at 5 mg/mL. This near-complete inhibition indicates that the extract contains potent COX-2 inhibitory constituents. In contrast, schaftoside alone showed relatively modest inhibition, with consistent values ranging from 85.2% to 86.9% across all tested concentrations (0.5–5 mg/mL), suggesting limited potency or a plateau effect when acting alone. These findings demonstrate that, while the crude extract effectively suppresses COX-2 activity at the enzymatic level, schaftoside alone does not account for the full inhibitory effect. It is therefore plausible that other bioactive constituents in *C. nutans* may contribute synergistically or independently to the COX-2 inhibition observed in the whole extract. In summary, the results provide strong evidence that *C. nutans* possesses potent COX-2 inhibitory activity, likely mediated by schaftoside in combination with additional co-existing compounds. This supports its traditional use as an anti-inflammatory agent and highlights its potential application in developing natural COX-2 inhibitors for therapeutic use.

#### 2.2.4. Downregulation of Inflammatory Gene Expression (Figure 4A–E)

To further elucidate the anti-inflammatory mechanisms, RT-PCR was performed to assess the mRNA levels of five key pro-inflammatory genes: *COX-2*, *iNOS*, *IL-6*, *TNF-α*, and *PGE*_2_
*EP*_2_. LPS stimulation markedly increased the expression of all these genes compared to untreated controls and sing Diclofenac sodium 25 μM dissolved in 0.1% DMSO as a positive control. Pre-treatment with *C. nutans* extract at 100 μg/mL significantly suppressed the expression of all five genes, bringing them nearly back to baseline. Schaftoside at 20 μg/mL showed a comparable effect, while lower doses—*C. nutans* extract at 50 μg/mL and schaftoside at 5 μg/mL—produced partial but measurable reductions. The gene expression data validate the protein-level findings and demonstrate multi-targeted anti-inflammatory action through the inhibition of both cytokines and enzymes. From this study, the results exhibited that *C. nutans* extract (≤250 μg/mL) and schaftoside (≤20 μg/mL) are non-cytotoxic. At the protein level, COX-2 and iNOS were significantly reduced in both pre- and post-treatment models. At the gene level, all key inflammatory genes (*COX-2*, *iNOS*, *IL-6*, *TNF-α*, and *PGE*_2_
*EP*_2_) were significantly downregulated. The most effective concentrations were *C. nutans* extract at 100 μg/mL and schaftoside at 20 μg/mL, which likely act through inhibition of NF-κB signaling, reducing pro-inflammatory mediator production. This comprehensive analysis confirms that both *C. nutans* extract and schaftoside exhibit potent, dose-dependent anti-inflammatory effects, working at both the transcriptional and translational levels. These results support their therapeutic potential in treating inflammation-driven diseases.

**Figure 4 ijms-26-06029-f004:**
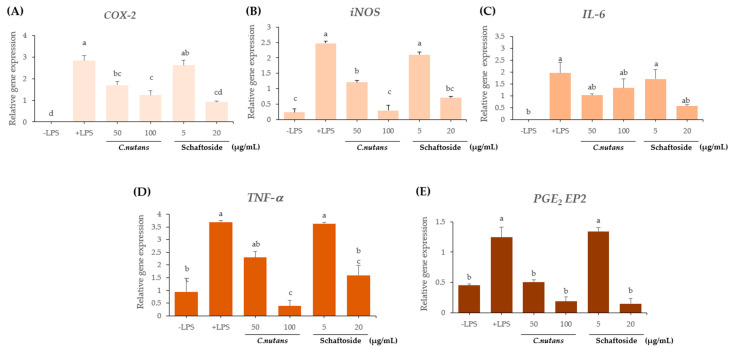
Effect of *C. nutans* extract and schaftoside on pro-inflammatory gene expression in LPS-stimulated RAW 264.7 cells. Macrophages were pre-treated with 95% ethanolic *C. nutans* extract or schaftoside for 1 h, followed by stimulation with 100 ng/mL LPS for 12 h. Total RNA was extracted, and RT-PCR was performed to assess the mRNA expression levels of pro-inflammatory genes: (**A**) *COX-2*, (**B**) *iNOS*, (**C**) *IL-6*, (**D**) *TNF-α*, and (**E**) *PGE*_2_
*EP2*. GAPDH was used as an internal control. A representative result from two independent experiments is shown. Gene expression was quantified and presented in arbitrary units (A.U.). Different lowercase letters indicate statistically significant differences between groups (*p* < 0.05).

### 2.3. Anti-Cell Death Activity of C. nutans Extract and Schaftoside Against LPS-Induced Endothelial Cell Death

*C. nutans* extract and its bioactive schaftoside were further investigated on lowering inflammation and inhibiting endothelial cell death. Cell death caused by LPS derived from gram-negative bacteria has been reported, particularly in sensitive cell types such as endothelial cells. Previously, we demonstrated the protective effect of *C. nutans* extract in lowering LPS-induced cell death in bovine endothelial cells, CCL-209 cells [20]. In this study, we compared the efficiency of the extract and its bioactive compound schaftoside. Cytotoxicity was assessed using a cell viability assay at concentrations ranging from 0.32 to 200 μg/mL for the extract and 10 to 40 μM for schaftoside. As shown in Figure 5A, both compounds exhibited low cytotoxicity; based on the dose-dependent cytotoxicity profile, concentrations below 200 µg/mL for *C. nutans* extract and below 40 µM for schaftoside were considered sublethal, with the highest non-toxic dose retaining 98% and 59.5% viability, respectively. These concentrations were thus selected for further analysis in LPS-induced cell death rescue assays.

To assess protection against LPS-induced cell death, CCL-209 cells were treated with 25 ng/mL LPS in the presence or absence of the selected concentrations of extract or schaftoside. LPS treatment alone drastically reduced cell viability to 9.25% at 24 h (Figure 5B). Co-treatment with *C. nutans* extract at 200 μg/mL fully restored cell viability to 100% at 24 h and 89.5% at 48 h, demonstrating strong protective effects. In comparison, schaftoside showed a moderate but dose-dependent protective effect. Treatment with 10, 20, and 40 μM schaftoside resulted in 4.25%, 10%, and 17.25% cell viability at 24 h and 35.5%, 37.75%, and 48.25% at 48 h, respectively. These findings suggest that while both *C. nutans* and schaftoside confer protection against LPS-induced endothelial damage, the crude extract exhibits superior efficacy.

### 2.4. Antiviral Activity C. nutans Extract and Schaftoside Against Herpes Simplex Virus (HSV)

The antiviral potential of *C. nutans* extract against HSV-2 was investigated through cytotoxicity testing and infection stage-specific inhibition assays. As shown in Figure 6A, *C. nutans* extract displayed low cytotoxicity, with a significant reduction in cell viability (to 80.19%) observed only at the highest concentration tested (2000 μg/mL). Based on this, sublethal concentrations (31.25–500 μg/mL) were used for further testing. The anti-HSV-2 activity of the extract was evaluated under two conditions: during infection and after infection. In the during-infection condition, where the extract was added concurrently with the virus, a dose-dependent inhibition of HSV-2 was observed (Figure 6B). Notably, at 500 μg/mL, the extract achieved 72.28% inhibition. In the after-infection condition, where the extract was added after viral adsorption, inhibition was less pronounced but still statistically significant, with 27.12% inhibition at 500 μg/mL (Figure 6C). These findings indicate that *C. nutans* extract exhibits strong anti-HSV-2 activity, particularly when present during the early stage of viral infection.

## 3. Discussion

Wound healing is a tightly regulated process comprising four overlapping phases: hemostasis, inflammation, proliferation, and remodeling. Each phase involves a cascade of cellular and molecular events orchestrated by immune cells, endothelial cells, fibroblasts, and keratinocytes to restore tissue integrity [1]. Disruptions in this sequence—especially during the inflammatory phase—can lead to delayed healing or chronic wounds [2].

In the present study, *C. nutans* extract and its major flavonoid component, schaftoside, demonstrated potent anti-inflammatory and endothelial-protective activities in vitro. While schaftoside was identified as a key bioactive flavonoid, the extract consistently exhibited stronger effects in multiple assays. This suggests potential synergism among minor flavonoids—such as orientin, isovitexin, and vitexin—and other phytochemicals that may enhance therapeutic efficacy. This synergy is supported by prior studies in herbal pharmacology, where multi-compound interactions have been shown to increase the bioavailability, stability, and bioactivity of plant-based formulations [4,9]. Schaftoside, in particular, has been reported to inhibit pro-inflammatory cytokines such as *TNF-α* and *IL-6*, which play key roles in both acute and chronic inflammation [17,18,19,20]. Orientin and isovitexin have shown strong free radical scavenging activity via DPPH and ABTS assays, highlighting their potential to protect tissues from oxidative stress-induced damage [9]. The combination of these bioactive flavonoids may result in a synergistic anti-inflammatory effect, whereby their pharmacological actions complement and amplify one another to produce a more potent therapeutic outcome than any single compound alone. This concept of phytochemical synergy is well supported in botanical pharmacology, particularly in complex pathological conditions such as chronic inflammation and infection, where modulation at multiple molecular targets is often more effective than single-agent interventions [4,9].

At the molecular level, *C. nutans* extract significantly downregulated the expression of key pro-inflammatory mediators, including COX-2, iNOS, TNF-α, IL-6, and PGE_2_, in LPS-stimulated RAW 264.7 macrophages. These molecules represent both upstream triggers and downstream amplifiers within the inflammatory cascade. TNF-α and IL-6 are early-phase cytokines, rapidly induced by TLR4 activation, which orchestrate leukocyte recruitment and systemic immune responses. iNOS, responsible for excessive nitric oxide production, contributes to oxidative tissue damage and persistent inflammation. COX-2, an inducible enzyme, catalyzes the biosynthesis of prostaglandins such as PGE_2_, a key lipid mediator in inflammation. Among the PGE_2_ four receptor subtypes (EP1–EP4), EP2 has emerged as the predominant pro-inflammatory effector, particularly in macrophages, epithelial cells, and fibroblasts. While EP1 and EP3 are associated with calcium mobilization and smooth muscle contraction, and EP4 with immunosuppression and resolution signaling, EP2 uniquely drives sustained inflammation via cAMP-PKA-CREB signaling. This promotes persistent transcription of *IL-6*, *TNF-α*, and *COX-2*, forming a feed-forward loop that amplifies and prolongs the inflammatory response. Moreover, EP2 signaling is known to impair macrophage phagocytic function and promote MMP expression and fibroblast proliferation—hallmarks of chronic inflammation, tissue degradation, and impaired wound resolution. The overactivation of EP2, rather than the other EP receptors, is therefore considered a central pathological driver in diseases such as rheumatoid arthritis, colitis, and fibrosis. The extract’s ability to suppress both gene transcription and protein expression under pre- and post-treatment conditions suggests it interferes at multiple key regulatory nodes, likely by inhibiting the NF-κB and MAPK pathways—major upstream regulators of iNOS, COX-2, and cytokine expression. Notably, COX-2 and TNF-α appeared especially sensitive to suppression, implying that *C. nutans* targets the initiation phase of inflammation while concurrently disrupting PGE_2_–EP2-mediated amplification in the later stages of the immune response [20,21,22]. Phytochemically, schaftoside—the major C-glycosyl flavone in *C. nutans*—has been reported to inhibit COX-2 and iNOS expression by blocking IκBα phosphorylation and reducing nuclear translocation of NF-κB p65, thereby diminishing the inflammatory output of macrophages [16,17,18]. However, in our study, schaftoside showed only moderate inhibitory activity, supporting the idea that the anti-inflammatory effects of the extract result from the synergistic interaction among schaftoside, orientin, vitexin, and other co-occurring flavonoids in the extract [5,6,7]. Interestingly, other studies have suggested that *C. nutans* may also modulate oxidative stress through activation of the Nuclear factor erythroid 2-related factor 2/Heme oxygenase-1 (Nrf2/HO-1) pathway, offering an additional layer of cytoprotection during inflammation [9]. Collectively, these findings support the hypothesis that *C. nutans* extract can attenuate inflammation through both transcriptional and enzymatic suppression of key mediators, making it a promising candidate for managing inflammation-associated wound complications.

In addition to these flavonoids, three chlorophyll-derived pheophytins were identified—132-hydroxy-(132-R)-phaeophytin b, 132-hydroxy-(132-S)-phaeophytin a, and 132-hydroxy-(132-R)-phaeophytin a—which are notable for their antiviral activity. These compounds are believed to interfere with the glycoproteins on the viral envelope of HSV-2, thereby blocking viral attachment and entry into host cells. Prior studies have demonstrated that pheophytins can integrate into lipid membranes and disrupt virus–host membrane fusion, a critical step in viral infection [6,12,23,24,25]. In our previous study, we demonstrated the antibacterial activity of *C. nutans* against key wound-associated pathogens, including *Staphylococcus epidermidis*, *Staphylococcus aureus*, *Pseudomonas aeruginosa*, and *Escherichia coli* [15]. Additional evidence from our recent study, supporting its antiviral properties, further reinforces the potential of *C. nutans* and schaftoside as a multifunctional wound-healing agent through both antibacterial and antiviral mechanisms.

Moreover, the extract exhibited cytoprotective effects in CCL-209 endothelial cells challenged with LPS. This effect is particularly relevant to wound healing, as endothelial cell survival is crucial for angiogenesis and nutrient delivery to regenerating tissues [2,8,9]. Endothelial-derived extracellular vesicles (EVs) can further promote fibroblast activation through transforming growth factor-beta (TGF-β) and platelet-derived growth factor (PDGF) signaling, leading to ECM deposition and granulation tissue formation. Additionally, under conditions of reduced TNF-α/IL-6 and elevated IL-10/TGF-β, fibroblasts become more responsive to regenerative signals, such as vascular endothelial growth factor (VEGF) and basic fibroblast growth factor (bFGF). VEGF facilitates endothelial proliferation and capillary formation, while bFGF promotes fibroblast expansion, ECM synthesis, and re-epithelialization, ultimately enhancing granulation and wound closure [26,27,28].

Despite these promising findings, this study has limitations. The biological activities were evaluated using in vitro macrophage and endothelial cell models, without direct investigation of fibroblast behavior, migration, or ECM production. In addition, no in vivo wound-healing models were employed. Therefore, while the extract exhibited favorable activities at the cellular and molecular levels, its functional relevance to tissue repair in living systems remains to be validated.

In conclusion, this study provides preliminary evidence that *C. nutans* extract and its flavonoid constituents exhibit multi-targeted activities relevant to wound healing, including suppression of pro-inflammatory cytokines (e.g., TNF-α, IL-6), inhibition of COX-2 activity, protection of endothelial cells, and interference with HSV-2 entry mechanisms (Figure 7). However, further studies—particularly in fibroblast-specific and in vivo wound models—are needed to confirm these findings and to elucidate the extract’s full therapeutic potential in wound management applications.

## 4. Materials and Methods

### 4.1. Plant Extraction

Fresh *C. nutans* leaves were collected from Lampang Province, Thailand, in January 2022. The species was authenticated by Asst. Prof. Dr. Angkana Inta, and voucher specimens (001–003) were deposited at the Department of Biology, Faculty of Science, Chiang Mai University. Leaves were washed with tap water, air-dried at room temperature, and cut into small pieces. The plant material was extracted with 95% ethanol (0.5 g/mL) and sonicated for 30 min. The extract was filtered through a 0.22 µm nylon membrane and diluted with 95% ethanol to a final concentration of 3000 µg/mL for HPLC analysis.

### 4.2. Analysis of Bioactive Compound Using HPLC-DAD

The schaftoside standard (Sigma-Aldrich, Merck KGaA, Darmstadt, Germany) was freshly prepared at 500 μg/mL in HPLC-grade methanol and serially diluted (0.78125–50 μg/mL) to construct a calibration curve. Chromatographic analysis was performed using an Agilent 1200 Series HPLC system (Agilent Technologies, Santa Clara, CA, USA) equipped with an Eclipse XDB-C18 column (4.6 × 150 mm, 5 μm) maintained at 35 °C. The mobile phase consisted of 0.1% formic acid in deionized water (solvent A) and acetonitrile (solvent B), 0 min (90:10), 5 min (86:14), 10 min (82:18), 15 min (78:22), delivered at a flow rate of 0.7 mL/min under gradient conditions. The injection volume was 10 μL, and detection was carried out at 340 nm. Linearity was confirmed with an R^2^ value of 0.9999. Precision was evaluated through five replicate injections of the extract (3000 μg/mL), with intraday and interday relative standard deviations (%RSDs) below 2%. Accuracy was assessed via spiked recovery experiments at 1500, 3000, and 6000 μg/mL, yielding recoveries within the ICH-acceptable range of 90–120%. Sensitivity was evaluated by determining the limit of detection (LOD = 0.5635 μg/mL) and the limit of quantification (LOQ = 1.7077 μg/mL) using the equations LOD = 3.3σ/S and LOQ = 10σ/S, where σ is the standard deviation of the response, and S is the slope of the calibration curve. These results indicate that the method is accurate, precise, sensitive, and suitable for routine quantification of schaftoside in botanical extracts.

### 4.3. The Effect of C. nutans and Schaftoside on Anti-Inflammation

#### 4.3.1. Reagents and Kits

COX-2, iNOS, and actin antibodies were obtained from Santa Cruz Biotechnology (Santa Cruz, CA, USA). Lipopolysaccharide (LPS; *Escherichia coli* O127:E8) was purchased from Sigma-Aldrich (St. Louis, MO, USA). Cell viability was assessed using the CellTiter 96^®^ AQueous One Solution Cell Proliferation Assay (Promega, Madison, WI, USA). Nitric oxide (NO) levels were measured using a colorimetric assay kit from Oxford Biomedical Research (Rochester Hills, MI, USA), and PGE_2_ levels were determined using the PGE_2_ EIA Monoclonal Kit (Cayman Chemical, Ann Arbor, MI, USA). Schaftoside was obtained from ChromaDex Inc. (Irvine, CA, USA), Sigma-Aldrich, and Indofine Chemical Company, Inc. (Hillsborough, NJ, USA). Unless otherwise specified, all other chemicals and reagents were purchased from Fisher Scientific (Pittsburgh, PA, USA).

#### 4.3.2. Cell Cultures

RAW 264.7 murine macrophage cells (ATCC, Manassas, VA, USA) were cultured in Dulbecco’s Modified Eagle’s Medium (DMEM) supplemented with 10% fetal bovine serum (FBS), 100 U/mL penicillin G, 100 μg/mL streptomycin, and 0.25 μg/mL amphotericin B. Cells were maintained at 37 °C in a humidified incubator with 5% CO_2_.

#### 4.3.3. Cell Proliferation Assay

The impact of *C. nutans* extract and its purified bioactive compound, schaftoside, on cell proliferation in RAW 264.7 macrophages was assessed using the CellTiter 96^®^ AQueous One Solution Cell Proliferation Assay (Promega, Madison, WI, USA). RAW 264.7 cells were seeded at a density of 1.0 × 10^5^ cells per well in 96-well tissue culture plates (*n* = 4) and allowed to adhere for 24 h. Cells were then treated with varying concentrations of *C. nutans* extract (100–500 μg/mL) or schaftoside (10–40 μg/mL) in medium containing 1% serum. After 24 h of exposure, 20 μL of the CellTiter 96 reagent was added to each well, followed by 1 h incubation at 37 °C. Cell viability was determined by measuring absorbance at 490 nm using a microplate reader (BioTek Instruments, Winooski, VT, USA).

#### 4.3.4. Western Blot Analysis

RAW 264.7 murine macrophage cells were seeded at a density of 1.0 × 10^6^ cells per well in 6 cm culture dishes and incubated for 24 h to achieve complete confluence prior to conducting pre- and post-treatment assays. For the pre-treatment protocol, cells were exposed to *C. nutans* extract at concentrations of 5 and 50 μg/mL or schaftoside at 5 and 20 μg/mL for 1 h in serum-free medium, using DMSO as the vehicle control. This was followed by stimulation with lipopolysaccharide (LPS) at 100 ng/mL for an additional 12 h. In the post-treatment protocol, cells were first stimulated with 100 ng/mL LPS for 1 h, after which they were treated with either the extract or schaftoside for 12 h. Following treatment, total cellular proteins were extracted using RIPA lysis buffer (1× PBS, 1% NP-40, 0.5% sodium deoxycholate, 0.1% SDS), supplemented with protease inhibitors (1 mM PMSF, 5 μg/mL aprotinin and leupeptin) and phosphatase inhibitors (1 mM Na_3_VO_4_, 1 mM NaF). Protein concentrations were quantified using the bicinchoninic acid (BCA) assay (Pierce, Rockford, IL, USA). Equal amounts of protein (50 μg) were separated via SDS-PAGE and transferred onto nitrocellulose membranes (Pall Life Sciences, Pensacola, FL, USA). Membranes were blocked with 5% skim milk in Tris-buffered saline containing 0.05% Tween-20 (TBS-T) for 1 h at room temperature and subsequently incubated with primary antibodies overnight at 4 °C. After three washes in TBS-T, membranes were incubated with horseradish peroxidase (HRP)-conjugated secondary antibodies for 1 h, followed by additional washes. Protein bands were detected using enhanced chemiluminescence (ECL) reagents (Thermo Scientific, Rockford, IL, USA).

#### 4.3.5. The Effect of *C. nutans* and Schaftoside as Human COX-2 Inhibitors

The inhibitory activity of chromone compounds against COX-2 was assessed using a commercial COX (human) inhibitor screening kit (Cat. No. 701080, Cayman Chemical, Ann Arbor, MI, USA), following the manufacturer’s instructions. Briefly, 10 μL of test compound was added to 160 μL of reaction buffer (0.1 M Tris-HCl, pH 8.0, containing 5 mM EDTA and 2 mM phenol), along with 10 μL of COX-2 enzyme and 10 μL of heme. After incubation at 37 °C for 10 min, the reaction was initiated by adding 10 μL of arachidonic acid, followed by immediate mixing and further incubation at 37 °C for 2 min. PGH_2_ was converted to PGF_2_α by adding 30 μL of stannous chloride and incubating at room temperature for 5 min. The amount of PGF_2_α produced was quantified by enzyme immunoassay using a 96-well plate pre-coated with mouse anti-rabbit monoclonal antibody. The plate was incubated with the reaction mixture for 18 h at room temperature, washed to remove unbound reagents, and Ellman’s reagent (containing the substrate for acetylcholinesterase) was added. The yellow color product from the enzymatic reaction was measured at 410 nm using a UV microplate reader. The percentage inhibition of COX-2 activity was calculated by comparing PGF_2_α levels in compound-treated samples to those in the control using the following equation:$$\mathrm{Inhibition}\;(\%)=\;\frac{[PGF2\alpha ]control-[PGF2\alpha ]sample}{{\left[PGF2\alpha \right]}_{control}}\;\times 100$$
where [PGF2α]control  =  PGF2α produced in untreated sample, and        [PGF2α ]sample =  PGF2α produced in compound-treated sample.

The assay was performed in triplicate wells, and celecoxib was used as a positive control. The test compounds, dissolved in 5% DMSO, were evaluated at a concentration of 30 µM.

#### 4.3.6. Reverse Transcription-Polymerase Chain Reaction (RT-PCR)

Total RNA was extracted using the Total RNA Kit (Omega Bio-Tek, Norcross, GA, USA) and reverse-transcribed into cDNA using the Verso cDNA Kit (Thermo Scientific), following the manufacturer’s instructions. PCR amplification was performed using ReadyMix Taq polymerase (Sigma-Aldrich), with specific primers targeting mouse COX-2, iNOS, IL-6, TNF-α, PGE2 (the EP2 receptor is a subtype of the PGE2 receptor family), and GAPDH, as listed in Table 1. The thermal cycling conditions were set at 94 °C for 30 s (denaturation), 55 °C for 30 s (annealing), and 72 °C for 1 min (extension). The number of amplification cycles was 20 for TNF-α; 25 for COX-2, IL-6, and GAPDH; and 28 for iNOS.

### 4.4. Anti-Cell Death Assay of C. nutans Extract and Schaftoside in CCL209 Cell Line

The CCL-209 (CPAE) endothelial cell line, derived from the main stem pulmonary artery of a young cow (Bos taurus), was cultured in DMEM supplemented with 20% fetal bovine serum (FBS), 100 U/mL penicillin G, and 100 μg/mL streptomycin. Vero cells, a monkey kidney cell line, were maintained in Minimal Essential Medium (MEM) supplemented with 10% FBS and the same antibiotic concentrations. All cells were incubated at 37 °C in a humidified atmosphere containing 5% CO_2_. To assess cytotoxicity, cell viability assays were performed in CCL-209 cells. Briefly, 7000 cells/well were seeded in 96-well plates one day prior to treatment. *C. nutans* extract and schaftoside were prepared at various concentrations in culture medium and added to the cells. Treated cells were incubated for 24 and 48 h, after which viability was assessed using PrestoBlue™ Cell Viability Reagent (Thermo Scientific, Rockford, IL, USA). The colorimetric change was measured at 570 nm, with 595 nm as the reference wavelength. The percentage of cell viability was calculated relative to untreated controls, which were set as 100%, using the following equation:% cell viability = [(OD_570_ − OD_595_) test/(OD_570_ − OD_595_) control] × 100

Sublethal doses (>85% viability) were used to assess protection against LPS-induced cytotoxicity. CCL-209 cells were seeded in 96-well plates and treated with 25 ng/mL LPS, with or without *C. nutans* extract or schaftoside. Cell viability was measured after 24 h using the PrestoBlue™ assay.

### 4.5. Anti-HSV Activity

Vero cells (80,000 cells/well) were seeded in 24-well plates one day prior to HSV-2 infection. For the during-infection condition, 50 PFU of HSV-2 was added, with or without *C. nutans* extract, and incubated for 1 h to allow viral adsorption. For the after-infection condition, cells were first infected with 50 PFU of HSV-2 for 1 h, followed by treatment with varying concentrations of the extract. In both conditions, cells were washed once and overlaid with 2% MEM containing carboxymethylcellulose (CMC), then incubated at 37 °C for 48 h. Plaques were stained with crystal violet, and the percentage of inhibition was calculated, relative to the untreated control (set at 100%).

## Figures and Tables

**Figure 1 ijms-26-06029-f001:**
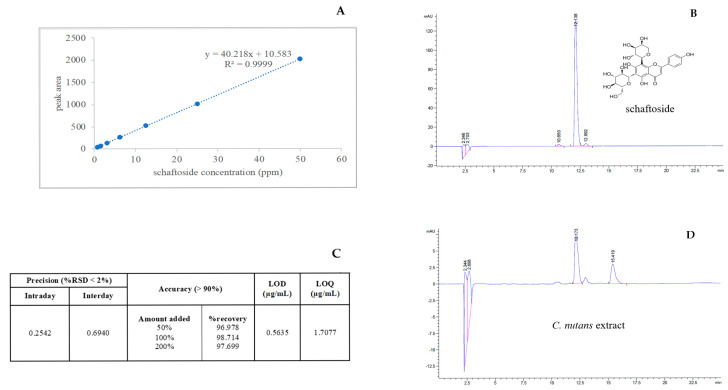
HPLC analysis and validation of schaftoside quantification. (**A**) Calibration curve of schaftoside over the range 0.78–50 μg/mL; (**B**) Representative chromatogram of schaftoside standard (100 μg/mL); (**C**) Summary of precision and recovery data; (**D**) Chromatogram of *C. nutans* ethanolic leaf extract.

**Figure 3 ijms-26-06029-f003:**
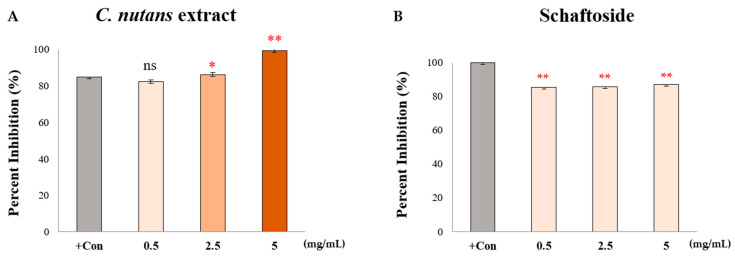
(**A**) *C. nutans* extract and (**B**) schaftoside inhibited human COX-2 activity, as determined using a COX (human) inhibitor screening kit (Cayman Chemical). Results are expressed as mean ± SD from three independent replicates. Asterisks indicate statistically significant differences compared to the positive control group (+Con: Diclofenac sodium 25 μM) (* *p* < 0.05, *** p* < 0.01); ns = not significant.

**Figure 5 ijms-26-06029-f005:**
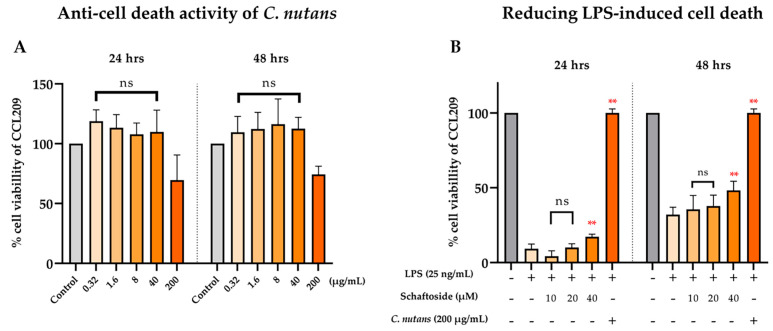
Anti-cell death activity of *C. nutans* extract and schaftoside. (**A**) Cell viability assay was performed to evaluate cytotoxicity of *C. nutans* extract (0.32–200 μg/mL) and its active component, schaftoside (10–40 μM), in bovine endothelial cells, CCL-209; (**B**) Sublethal doses were selected to investigate the effect of *C. nutans* extract and schaftoside on reducing LPS-induced cell death in CCL-209. The cells were treated with LPS at the concentration of 25 ng/mL in the presence or absence of *C. nutans* extract and schaftoside, where the control was 2% DMSO in media. The cell viability was calculated relative to that of the non-treatment control, which was set as 100%. Asterisks indicate statistically significant differences compared to the LPS-only group (** *p* < 0.01)*;* ns = not significant.

**Figure 6 ijms-26-06029-f006:**
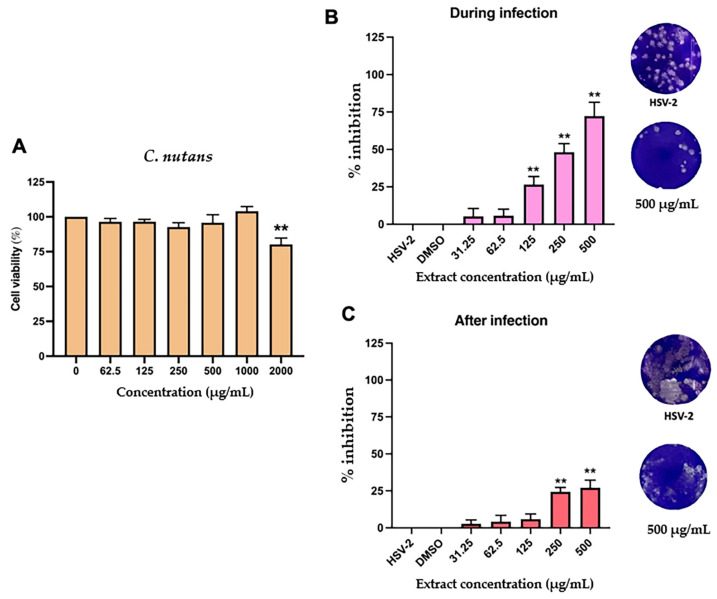
Anti-HSV-2 activity of *C. nutans* extract in Vero cells. (**A**) Cytotoxicity was assessed at 62.5–2000 μg/mL; only the highest dose (2000 μg/mL) significantly reduced viability to 80.19% (*p* < 0.01). (**B**) During-infection assay showed dose-dependent HSV-2 inhibition, reaching 72.28% at 500 μg/mL (*p* < 0.01). (**C**) Post-infection treatment yielded lower but significant inhibition (27.12% at 500 μg/mL; *p* < 0.01). Plaque images illustrate viral suppression at 500 μg/mL. Asterisks indicate statistically significant differences compared to control group (** *p* < 0.01).

**Figure 7 ijms-26-06029-f007:**
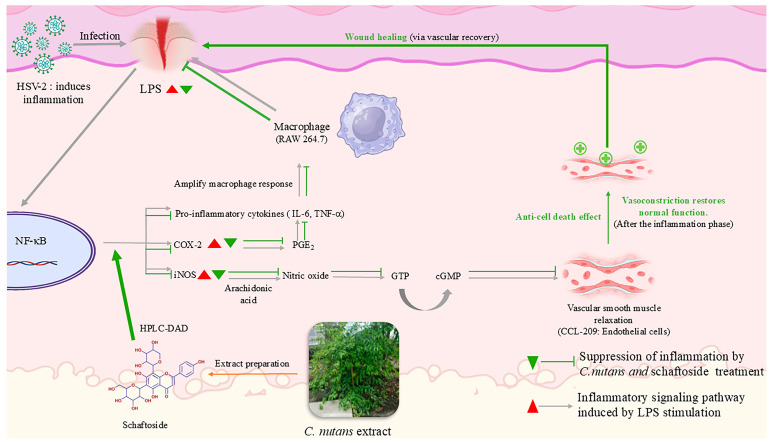
Schematic representation of the wound-healing mechanism modulated by *C. nutans* extract and schaftoside. HSV-2 infection or LPS stimulation activates macrophages (RAW 264.7), triggering NF-κB signaling and upregulation of pro-inflammatory mediators, including COX-2, iNOS, IL-6, TNF-α, and PGE_2_. These mediators amplify inflammation and contribute to endothelial dysfunction. *C. nutans* extract and schaftoside inhibit this pathway by suppressing NF-κB activation, reducing inflammatory gene expression, and inhibiting COX-2 activity. Additionally, the extract exhibits anti-cell death effects in CCL-209 endothelial cells and suppresses HSV-2 infection, supporting vascular function and preventing infection-related delays. These actions collectively promote wound healing.

**Table 1 ijms-26-06029-t001:** Specific primer sequences of inflammatory genes for RT-PCR.

Genes	Primer Sequences	
	Sense Strand (5′–3′)	Anti-Sense Strand (5′–3′)
*COX-2*	CCCCCACAGTCAAAGACACT	GAGTCCATGTTCCAGGAGGA
*iNOS*	GTCTTGCAAGCTGATGGTC	CATGATGGTCACATTCTGC
*IL-6*	CCGGAGAGGAGACTTCACAG	GGAAATTGGGGTAGGAAGGA
*TNF-α*	CGTCAGCCGATTTGCTATCT	CGGACTCCGCAAAGTCTAAG
*PGE*_2_ *EP*_2_	GTGGCCCTGGCTCCCGAAAGTC	GGCAAGGAGCATATGGCGAAGGTG
*GAPDH*	CAGGAGCGAGACCCCACTAACAT	GTCAGATCCACGACGGACACATT

## Data Availability

The datasets used and analyzed during the current study are available from the corresponding author on reasonable request.

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
