# Peer review of "Integrative Wound-Healing Effects of Clinacanthus nutans Extract and Schaftoside Through Anti-Inflammatory, Endothelial-Protective, and Antiviral Mechanisms"

_ijms, 2025, doi:10.3390/ijms26136029_

Round 1
Reviewer 1 Report
Comments and Suggestions for Authors
- There is a syntax error in page 2 line 81.
- The paper mainly focuses on anti-inflammatory research. The introduction could be shorter. The vasculoprotective effects can be briefly mentioned. Merge the last two paragraphs of the introduction section and streamline them.
- The 2.1 and 4.2 section is a conventional analytical method. It Can be included in supplementary materials. Only the content data needs to be retained. Pay attention to significant figures in line 152 and 153.
- Figure 3 lacks differential analysis. Clarify the meaning of +con in Figure 3. The format and position of A and B in all figures should be consistent.
- Figure 5 should clearly indicate the groups for comparison in the analysis of significant differences.
- Figure 6 ml should be mL. The font size of the words in B and C should be the same.
- The discussion section should be improved.
Author Response
Reviewer 1
Comments 1: There is a syntax error in page 2 line 81.
Response 1: The syntax error in Page 2, Line 81 has been corrected.
Comments 2: Paper mainly focuses on anti-inflammatory research. The introduction could be shorter. The vasculoprotective effects can be briefly mentioned. Merge the last two paragraphs of the introduction section and streamline them.
Response 2: The introduction has been shortened to emphasize the anti-inflammatory activity. The vasculoprotective and antiviral effects were briefly mentioned and de-emphasized. The last two paragraphs of the introduction were merged and streamlined for clarity and conciseness.
Comments 3: The 2.1 and 4.2 section is a conventional analytical method. It can be included in supplementary materials. Only the content data needs to be retained. Pay attention to significant figures in line 152 and 153.
Response 3: Sections 2.1 and 4.2 were revised to retain only the essential content data. The methodological details have been moved to the supplementary materials. The number of significant figures in Lines 152 and 153 has been adjusted accordingly.
Comments 4: Figure 3 lacks differential analysis. Clarify the meaning of +con in Figure 3. The format and position of A and B in all figures should be consistent.
Response 4: Differential analysis has been added to Figure 3. The meaning of “+con” has been clarified and described in the Methods section. The placement and formatting of panels A and B have been standardized across all figures.
Comments 5: Figure 5 should clearly indicate the groups for comparison in the analysis of significant differences.
Response 5: The comparison groups in Figure 5 are now clearly indicated to enhance interpretability of the statistical analysis.
Comments 6: Figure 6 ml should be mL. The font size of the words in B and C should be the same.
Response 6: All instances of “ml” have been corrected to “mL.” The font size in panels B
Comments 7: The discussion section should be improved.
Response 7: The Discussion section has been thoroughly rewritten to improve clarity, depth of interpretation, and alignment with the study’s findings.

Reviewer 2 Report
Comments and Suggestions for Authors
The authors present an interesting and relevant article investigating the potential wound healing properties of Clinacanthus nutans extract and its major compound, schaftoside. However, I do have several critical comments that should be addressed before considering publication:
- Relevance of antiviral activity:
While the authors emphasize the antiviral properties of the extract, especially against HSV-2, I question the relevance of this parameter in the context of general wound healing. Unless viral infections (e.g., HSV-related ulcers) are explicitly targeted, the antiviral assays seem less relevant. In wound care, bacterial infections are far more common and impactful. Therefore, I suggest that antibacterial properties be assessed, or at the very least, that the emphasis on antiviral mechanisms in the conclusion be toned down. - Lack of data on fibroblast activity:
Fibroblasts are key cells in wound healing, especially during the proliferation and remodeling phases. However, the authors do not include any assays or discussion related to fibroblast migration, proliferation, or collagen production. Including such data, or discussing the potential effects of the extract on fibroblasts based on existing literature, would strengthen the translational relevance of the findings. - Clarity issues in the manuscript:
- In line 81, the sentence appears to be incomplete or grammatically incorrect: “, which are essential for viral attachment, membrane fusion, and host cell entry.” Please revise for clarity.
- In Figure 3, although the extract shows strong inhibition of COX-2 activity, schaftoside shows only a modest and non-dose-dependent inhibition. The conclusion that schaftoside is the main active component should be more cautiously worded, as the data suggest additional compounds in the extract may be responsible for the higher overall effect (potential synergism).
- A considerable number of abbreviations are used throughout the manuscript, but not all are included in the list at the end (e.g., RAW 264.7, LPS, DMSO, CMC, DMEM, ECL, HRP). Please ensure that all abbreviations are:
- Defined at first use in the main text, immediately followed by the abbreviation in parentheses.
- Used consistently throughout the manuscript after initial definition.
- This is particularly important for abbreviations such as CN (for C. nutans extract) and SCF (for schaftoside), which are introduced without prior explanation only late in the results section (e.g., Figures 2–4). These abbreviations should be clearly introduced when first mentioned in the methods or results section, to ensure clarity for the reader.
- Minor language and stylistic issues:
- Occasionally, sentence construction is awkward or inconsistent (e.g., switching between “C. nutans” and “CN”; sometimes “extract”, sometimes “compound”). I recommend a thorough language check or professional proofreading.
- Strength of the conclusion (Line 370)
Based on the above comments and the data presented, the conclusion drawn by the authors is too strong and should be rephrased to better reflect the preliminary nature of the findings. While the data support the potential of C. nutans extract in modulating inflammation and protecting endothelial cells, further validation, especially in relevant wound healing models (e.g., involving fibroblasts or in vivo studies), is required before these findings can be translated into therapeutic applications. - Suggestions for Improvement:
- The discussion should also explore the limitations of the study, such as the lack of in vivo validation or absence of fibroblast-based models.
Author Response
Reviewer 2
Comment 1: Relevance of antiviral activity: While the authors emphasize the antiviral properties of the extract, especially against HSV-2, I question the relevance of this parameter in the context of general wound healing. Unless viral infections (e.g., HSV-related ulcers) are explicitly targeted, the antiviral assays seem less relevant. In wound care, bacterial infections are far more common and impactful. Therefore, I suggest that antibacterial properties be assessed, or at the very least, that the emphasis on antiviral mechanisms in the conclusion be toned down.
Response 1: We acknowledge the reviewer’s concern regarding the relevance of antiviral activity in the context of general wound healing. The antibacterial activity of Clinacanthus nutans was previously established in our earlier study (Reference No. 15), and this has now been clearly highlighted in the Discussion section (Lines 342–348). In response to the comment, we have reduced the emphasis on antiviral mechanisms in both the Introduction and Conclusion to better align with the primary focus on antibacterial contributions to wound healing.
Comment 2: Lack of data on fibroblast activity: Fibroblasts are key cells in wound healing, especially during the proliferation and remodeling phases. However, the authors do not include any assays or discussion related to fibroblast migration, proliferation, or collagen production. Including such data, or discussing the potential effects of the extract on fibroblasts based on existing literature, would strengthen the translational relevance of the findings.
Response 2: We appreciate the reviewer’s insightful comment regarding the role of fibroblasts in wound healing. In response, we have acknowledged this limitation in the Conclusion section (Lines 371–373), noting that future studies should investigate the effects of C. nutans on fibroblast activity to further support its therapeutic potential in wound repair.
Comment 3: Clarity issues in the manuscript: In line 81, the sentence appears to be incomplete or grammatically incorrect: “, which are essential for viral attachment, membrane fusion, and host cell entry.” Please revise for clarity. In Figure 3, although the extract shows strong inhibition of COX-2 activity, schaftoside shows only a modest and non-dose-dependent inhibition. The conclusion that schaftoside is the main active component should be more cautiously worded, as the data suggest additional compounds in the extract may be responsible for the higher overall effect (potential synergism).
Response 3: We thank the reviewer for this valuable observation. The sentence in Line 81 has been revised for clarity and grammatical accuracy. Additionally, we have addressed the modest and non-dose-dependent effect of schaftoside by discussing the potential synergistic interactions between schaftoside and other flavonoids in the extract. This has been incorporated into the Discussion section (Lines 277–295) to more accurately reflect the complexity of the extract’s bioactivity.
Comment 4: considerable number of abbreviations are used throughout the manuscript, but not all are included in the list at the end (e.g., RAW 264.7, LPS, DMSO, CMC, DMEM, ECL, HRP). Please ensure that all abbreviations are: Defined at first use in the main text, immediately followed by the abbreviation in parentheses. Used consistently throughout the manuscript after initial definition. This is particularly important for abbreviations such as CN (for C. nutans extract) and SCF (for schaftoside), which are introduced without prior explanation only late in the results section (e.g., Figures 2–4). These abbreviations should be clearly introduced when first mentioned in the methods or results section, to ensure clarity for the reader.
Response 4: All abbreviations, including have now been clearly defined upon first use in the main text. To improve clarity and readability, we have replaced “CN” with C. nutans and “SCF” with schaftoside throughout the manuscript. We have also ensured that all abbreviations are used consistently throughout the manuscript and included in the abbreviation list at the end for clarity.
Comment 5: Minor language and stylistic issues: Occasionally, sentence construction is awkward or inconsistent (e.g., switching between “C. nutans” and “CN”; sometimes “extract”, sometimes “compound”). I recommend a thorough language check or professional proofreading.
Response 5: The manuscript has been thoroughly proofread by a native English-speaking editor to address all language and stylistic inconsistencies. We have ensured consistent terminology throughout, using C. nutans to refer to the extract and distinguishing it clearly from individual compounds such as schaftoside.
Comment 6: Strength of the conclusion (Line 370) Based on the above comments and the data presented, the conclusion drawn by the authors is too strong and should be rephrased to better reflect the preliminary nature of the findings. While the data support the potential of C. nutans extract in modulating inflammation and protecting endothelial cells, further validation, especially in relevant wound healing models (e.g., involving fibroblasts or in vivo studies), is required before these findings can be translated into therapeutic applications.
Response 6: We acknowledge the reviewer’s comment and have revised the conclusion to more accurately reflect the preliminary nature of our findings. The revised conclusion now emphasizes the potential of C. nutans extract in modulating inflammation and protecting endothelial cells, while also highlighting the need for further validation in fibroblast-based and in vivo wound healing models before advancing toward therapeutic application.
Comment 7: Suggestions for Improvement: The discussion should also explore the limitations of the study, such as the lack of in vivo validation or absence of fibroblast-based models.
Response 7: The Discussion section has been revised to include a clear acknowledgment of the study’s limitations, specifically the lack of in vivo validation and the absence of fibroblast-based models. These points are now discussed to provide a balanced interpretation of the findings and to guide directions for future research.

Round 2
Reviewer 1 Report
Comments and Suggestions for Authors
No
Author Response
Thank you so much for your help.
Reviewer 2 Report
Comments and Suggestions for Authors
Thank you for addressing my comments and implementing the requested changes. The revised manuscript shows clear improvements. Apart from the fact that not all abbreviations are yet included in the list, I have no further remarks.
Author Response
All the abbreviations were listed. Thank you so very much for your help and support.